# An analysis of deficiencies in the ethics committee data of certain interventional trials registered with the Clinical Trials Registry–India

Indraneel Chakraborty[1], Adya Shreya[1], Jaishree Mendiratta[1], Anant Bhan[2], Gayatri Saberwal[1] *

**1** Institute of Bioinformatics and Applied Biotechnology, Bengaluru, India, **2** Centre for Ethics, Yenepoya (deemed to be University), Mangaluru, India

* gayatri@ibab.ac.in

## Abstract

There is widespread agreement that clinical trials should be registered in a public registry, preferably before the trial commences. It is also important that details of each trial in the public record are complete and accurate. In this study, we examined the trial sites and ethics committee (EC) data for 1359 recent Phase 2 or Phase 3 interventional trials registered with Clinical Trials Registry–India (CTRI), to identify categories of problems that prevent the clear identification of which EC approved a given site. We created an SQLite database that hosted the relevant CTRI records, and queried this database, as needed. We identified two broad categories of problems: those pertaining to the understanding of an individual trial and those to adopting a data analytics approach for a large number of trials. Overall, about 30 problems were identified, such as an EC not being listed; an uninformative name of the EC that precluded its clear identification; ambiguity in which EC supervised a particular site; repetition of a site or an EC; the use of a given acronym for different organizations; site name not clearly listed, etc. The large number of problems with the data in the EC or site field creates a challenge to link particular sites with particular ECs, especially if a programme is used to find the matches. We make a few suggestions on how the situation could be improved. Most importantly, list the EC registration number for each EC, merge the site and EC tables so that it is clear which EC is linked to which site; and implement logic rules that would prevent a trial from being registered unless certain conditions were met. This will raise user confidence in CTRI EC data, and enable data based public policy and inferences. This will also contribute to increased transparency, and trust, in clinical trials, and their oversight, in India.

## Introduction

Clinical trial registries hold many details pertaining to each registered trial. Since registries in several countries now require that trials are registered prospectively [1], this provides a

(https://osf.io/t3mv5). All other files are available as Supporting Information files.

**Funding:** This work was supported by internal funding of the Institute of Bioinformatics and Applied Biotechnology, from the Department of Electronics, IT, BT and S&T of the Government of Karnataka. The funder had no role in the study design, data collection and analysis, interpretation of data, decision to publish, or preparation of the manuscript. No additional external funding was received for this study.

**Competing interests:** The authors have read the journal's policy and have the following competing interests: AB is a section editor of PLOS Global Public Health. GS has acted as a consultant to Mr. Dinesh Thakur on the various ways in which the workings of CTRI could be improved. The authors declare that they have no other competing interests. This does not alter our adherence to PLOS ONE policies on sharing data and materials.

permanent record of what was planned when the study started. Such records can then be compared with subsequent publications, to determine whether there were any discrepancies in the reporting of protocols or outcomes, for instance. Further, registry data have been put to many other uses, such as investigating the ethics, science, regulations, economics and so on of the clinical trials enterprise [2]. It is therefore important that at registration, details of each study are as complete and accurate as possible.

As mentioned, one of the important goals of a registry is to provide the public information about each registered trial. Repeatedly, academics and others [3,4], including the Science and Technology Committee of the UK House of Commons [5] have called for audits of trial data. Over the years, many researchers have performed limited audits, and have identified gaps and errors in registries [6–10] or between registries [11], identified discrepancies between the data in registries and in publications [12–14], investigated whether trial results were reported on time [15]; and so on. Some of these audits have been in 'real time' [16], or in 'ongoing' mode [17].

Coming to India, the Clinical Trials Registry–India (CTRI) was established in 2007, and on 15 July 2021, it had 34,859 records. In terms of the types of records held by the registry, (a) in a CTRI record, the field *Type of trial* has a dropdown option with four options, ie. Observational, Interventional, PMS (post-marketing study) and BA/BE (ie, Bioavailability/Bioequivalence studies). So, these four categories of trials are held by the registry. (b) aside from trials in Western medicine, a large number of trials in 'alternate' streams of medicine, ie ayurveda, yoga, unani, siddha and homeopathy (generally known as AYUSH) are conducted in India, which may also be registered with CTRI.

The CTRI website hosts a document that describes each field in the database [18]. In this, the ethics field has the following description: "Name of Ethics Committee (EC) and approval status: Provide name of EC from whom approval has been sought; for multi-center trials, add names of all ECs from whom approval has been sought. . ." The intention of this description is that it should be possible for the public to determine which committee gave approval to a given study at a given site. It is important to have accurate data pertaining to which ECs are linked to a given trial that has taken place, or is ongoing, in the country. An EC itself may be over-burdened with trial-related work [19], or physician members of the EC in their individual capacity may have high clinical burdens. Accurate EC data will help determine the frequency with which particular ECs supervise studies, and can quantify the load on each EC, or its physician members, at a given time. It would also be of great assistance in case of any mishap, when public health advocates will be able to immediately identify which EC needs to be contacted, instead of waiting for information from the government or the sponsor. It is known that researchers or health advocates may have to struggle to obtain information in a timely manner, as demonstrated by the Tamiflu case when researchers had a long struggle to obtain the clinical study reports from the sponsor or the regulator [20]. The greater the transparency around the ECs linked to a given trial, the more likely it is that the public will obtain rapid and detailed information in case a trial goes wrong.

By way of background, the Ethics Committee (EC) or Institutional Ethics Committee (IEC) in India is equivalent to the Research Ethics Committee (REC) or Institutional Review Board (IRB) in other countries. Here we briefly outline the steps typically required by local regulations for the registration and ethical assessment of projects until their approval:

a. Typically, projects need to be carried out with institutional support.

b. After a technical assessment (which could also be part of the ethics review process in some institutions) projects are submitted to the institutional ethics committee of the relevant institution(s) for approval.

c. Clinical trial projects also need to be registered with clinical trial registries, especially the Clinical Trials Registry–India, prior to recruitment of the first participant into the study.

d. For regulatory studies, permission is also needed from CDSCO.

e. For collaborative research involving international partners, permission is required from the Health Ministry Screening Committee.

f. For specific kinds of studies, additional permissions might be required—for example, for gene therapy research, approval from the Gene Therapy Advisory and Evaluation Committee would be required; for samples being shipped to other institutions, Material Transfer Agreements should be in place and special permissions might be required for export or import of samples; for studies involving handling of microorganisms or genetically engineered organisms, permissions from Institutional Biosafety Committees and other relevant bodies such as Review Committee on Genetic Manipulation might be required.

So, typically institutions have their own EC, and researchers are expected to get approval of their own institution's EC to participate in a trial. It is possible to have multicentre review, where the EC of one site has oversight of all the trial sites, but that would be allowed only for low-risk, non-regulatory trials, ie not the phase 2 and phase 3 trials of this study. We also need to be aware of the regulatory updates of the Government of India. In March 2019, the Ministry of Health and Family Welfare issued the New Drugs and Clinical Trials Rules [21]. These rules permitted a trial to be conducted at a site that lacked its own EC. In such cases, an EC from another institution, or an independent EC, may review the trial provided the EC is located in the same city as the site or is within a 50 km radius. These rules came into effect on 25 March 2019.

Does each trial record enable readers to determine which EC was linked to a particular site? In earlier work [9], as part of a larger study, we determined that (i) it was not always possible to identify each EC unambiguously; (ii) it was not always possible to determine which EC had approved a given site; and (iii) sometimes, more ECs were listed than sites for a particular study. In that work, we had merely provided examples of (i) and (ii), although we quantified (iii). Here, we wished to perform a more detailed audit of the EC data in CTRI, and identify any other lacunae with this data.

## Materials and methods

CTRI records were available at http://ctri.nic.in/Clinicaltrials/advancesearchmain.php. We downloaded the records of interest on 19 February 2021, when CTRI hosted 31,423 trials. Study records were available in an HTML format, and a sample record is provided as S1 File.

We developed an in-house script in R programming language (S2 File, available at https://osf.io/nmhd8/), which web scraped data from the CTRI records, and cleaned and processed the data for improved usability. The script then stored these data in a standardized SQLite database which was used for further analyses. This database is available at https://osf.io/t3mv5/ as S3 File. The schema of the database is available in S4 File. Further details of Methodology are available in S5 File. In the following sections, field names are italicized. Also, we use the terms trial, study and case interchangeably.

We wished to examine trials registered in recent years. There were 24,445 trials registered between 1 June 2016 and 19 February 2021, an almost five-year period, and we went on to identify the trials of interest in this set. 17,342 were interventional trials, as identified by the field *Type of trial*. Of these, 3160 were Phase 2, and 3200 were Phase 3, as identified by the field *Phase*. Phase 2 studies included Phase 2/Phase 3 trials, and Phase 3 studies included both

**Table 1. Number of trials at the different steps of identifying the cases of interest.**

|     | Step | Number of trials | Supplementary File |
| --- | --- | --- | --- |
| 1. | Trials registered 1 June 2016 onward | 24,445 | S6 File |
| 2. | Trials registered 1 June 2016 onward AND interventional | 17,342 | |
| 3. | Trials registered 1 June 2016 onward AND interventional AND Phase 2 | 3160 | |
| 4. | Trials registered 1 June 2016 onward AND interventional AND Phase 3 | 3200 | |
| 5. | Trials registered 1 June 2016 onward AND interventional AND Phase 2 AND Completed or Closed to recruitment | 736 | |
| 6. | Trials registered 1 June 2016 onward AND interventional AND Phase 3 AND Completed or Closed to recruitment | 840 | |
| 7. | 736 Phase 2 trials | 599 single site 137 multi-sites | S7 File |
| 8. | 840 Phase 3 trials | 582 single site 258 multi-sites | |
| 9. | Merging data from Phase 2 and Phase 3 trials, and removing duplicates | 1359 trials: 1012 single site 347 multi-sites | |

Phase 2/Phase 3 and Phase 3/Phase 4 trials. Of these, 736 and 840 studies in the Phase 2 and Phase 3 sets, respectively, were either Completed or Closed to recruitment, as identified by the field *Recruitment Status of Trial (India)*. The 736 Phase 2 trials had 599 single-site and 137 multi-site studies, and the 840 Phase 3 trials had 582 single-site and 258 multi-site studies. We merged the Phase 2 and Phase 3 data, and removed the 217 Phase 2/Phase 3 duplicates, to obtain 1359 trials, of which 1012 were single-site and 347 multi-site trials (Table 1).

For the unique set of 1359 trials, we went on to identify all those for which (i) no EC was listed; (ii) the number of sites exceeded the number of ECs; (iii) the number of ECs exceeded the number of sites; and (iv) the number of sites was the same as the number of ECs. In the last set, we identified cases where (a) there was just one site and one EC, and (b) the number of sites and ECs exceeded one. For sets (ii), (iii) and (iv b), we used a programme to find the best possible match of each trial site with an EC, and then manually checked all the site-EC matches to confirm whether the matches appeared to be correct. Further details of methodology are available in S5 File.

## Results

The studies were first divided into the following four categories: Those for which no ECs were listed (three cases); those with more sites than ECs (53 cases); those with more ECs than sites (38 cases) and those with an equal number of ECs and sites (1265 cases) (Table 2).

We first describe the problems with EC data, and with creating site-EC matches, for the first three subsets.

### Three trials for which no ECs were listed

There were three trials for which no EC was listed (S8 File). For these studies, no further analysis could be done.

**Table 2. Breakup of the set of 1359 Phase 2 or Phase 3 trials.**

|     | Result | Number of trials | File name |
| --- | --- | --- | --- |
| 1. | Trials for which no ECs were listed | 3 | S8 File |
| 2. | Trials with more sites than ECs | 53 | |
| 3. | Trials with more ECs than sites | 38 | |
| 4. | Trials with an equal number of ECs and sites | 1265 | |

## 53 trials which listed more sites than ECs

There were 53 trials in which the number of sites exceeded the number of ECs (S8 File). These studies had the following actual or possible problems with their site and EC data: (i) There were several cases of a site being repeated, with a single listing of the correct EC. (ii) There was one case with 39 sites, but a single EC. (iii) For many trials, some sites didn't have obvious ECs. (iv) There may have been one generically named 'Institutional Ethics Committee' for multiple sites. (v) In some cases the EC had a doctor's name linked to it, but it was not possible to clearly link such an EC to the site name. (vi) There was a study with two sites that had the same acronym. The correct site-EC match could be established only because one of the ECs also listed the city. (vii) One study had multiple sites with the generically named 'Government Medical College and Hospitals', requiring extra steps to find the correct matching EC. (viii) In one case the *Name of site* was the name of the department.

## 38 trials which listed more ECs than sites

There were 38 studies in which the number of ECs exceeded the number of sites (S8 File). For the purpose of analysis, this set was broken up into 22 single-site and 16 multi-site trials.

The 22 sites had a total of 50 ECs (S8 File). These studies had the following actual or possible problems with their site and EC data: (i) Although there was a single site, a given EC may have been repeated twice or thrice. In some trials, the repeated EC had a generic Institutional Ethics Committee (IEC) name. In other cases, it was not always clear whether the listed EC was linked to the site. (ii) There were cases where although there was only one site, there were ECs from two or more organizations. In some of these, one of the ECs was clearly linked to the site, but in others it was unclear which of the ECs was so linked. (iii) In one study, although it appeared that two ECs were listed, one of them was just a message ("Conditional approval by Institutional Ethics Committee was given on 29/10/17 which is six month before enrolment of first patient. So please consider this"). (iv) There was a case where the site names were listed in place of an EC. Although there were multiple problems with EC data in this dataset, we identified one trial where the registration number of the EC (the ECR) with the Central Drugs Standard Control Organisation (CDSCO) was listed, in addition to the name of the EC.

For the set of 16 studies, we performed a quantitative assessment. These trials had a total of 161 sites. Of these, we could not identify the EC linked to 23 (14%) sites, which were the locations of eight of the 16 trials (S8 File).

## Trials which listed an equal number of ECs and sites

The set of 1265 cases was divided into three groups: Where the number of sites and ECs was (a) one each (987 trials); (b) greater than one, and there was no particular difficulty in matching sites and ECs using a programme (236 trials); (c) greater than one, and there was considerable difficulty in matching sites and ECs using a programme (42 trials) (Table 3).

**Table 3. The breakup of the set of 1265 trials that had an equal number of sites and ECs.**

| | Result | Number of trials | File name |
|---|---|---|---|
| 1. | Trials with a single site and single EC | 987 | S9 File |
| 2. | Trials in which the number of sites equaled the number of ECs, which was greater than one: Simple cases. | 236 | |
| 3. | Trials in which the number of sites equaled the number of ECs, which was greater than one: Complicated cases. | 42 | |

## 987 trials in which there was a single site and single EC

There were 987 single-site studies with a single EC (S9 File). The categories of problems with this data are as follows:

(i) The *Name of the Committee* field–that is the name of the EC–may merely have listed the site's name. (ii) Generic names (such as 'institutional ethics committee') of the EC were used quite often. The name of the institution may, or may not, have been listed in this field as well. (iii) Some entries appeared to be casual references (such as 'ethical committee of Council') to particular ECs, rather than their proper names. (iv) A given institute may have listed its EC in different ways in different studies, leaving it unclear as to whether the same EC was being referred to. (v) Often, the name of the EC included an acronym. Usually, the acronym would be spelled out somewhere else in the trial record. However, we identified one case where it was not spelled out anywhere in the record. (vi) The same acronym may have referred to different organizations, in different trials. (vi) Information in the *Sites of Study* field may have been incomplete. Usually, if the name of the organization was unavailable in the *Name of Site*, it was available in the *Site Address*. However, it was not always provided anywhere in the *Sites of Study* field, although present elsewhere in the document. (vii) An independent EC may, or may not, have been identified as such. (viii) In a few cases, what appears to be the EC file number was listed instead of the name of the EC. One problematic situation was caused by five trials that listed the same EC file number.

As enumerated above, there were several problems with EC data in this dataset. Nevertheless, we identified four trials (CTRI/2017/01/007686, CTRI/2018/01/011488, CTRI/2018/06/014541, and CTRI/2019/01/017169), where the ECR was listed (ECR/262/Inst/UP/2013, ECR/541/Inst/KA/2014, ECR/346/Inst/AP/2013/RR-16, and ECR/400/Inst/Py/2013 respectively). This information was listed in addition to the name of the EC, in all except one case which only listed the ECR.

## 236 trials in which the number of sites equaled the number of ECs, which was greater than one

There were 236 trials in which the number of sites was equal to the number of ECs, with each being greater than one (S9 File). There was no particular difficulty in matching site and EC, where such matches existed, using a programme. For this set, we performed a quantitative assessment, based on which this set of studies could be broken down into three categories (Table 4). The EC could be unambiguously determined for all the sites of only 40% of the trials. For another 28%, the EC-site linkage could be determined because it was the last EC after all other ECs had been assigned to other sites. For 32% of the studies, there were two or more sites for which the EC could not be unambiguously determined.

For trials with two or three sites, sometimes no EC could be assigned to any of them. That is, 100% of ECs could not be assigned. For studies with 10 or more sites, up to 46% of ECs could not be assigned.

**Table 4. The breakup of the set of 236 trials in which the number of sites equaled the number of ECs, each greater than 1.**

|  | Result | Number of trials (percentage) |
|---|---|---|
| 1. | Trials in which all the site-EC matches were clear | 94 (39.8%) |
| 2. | Trials with one unclear site-EC match | 67 (28.4%) |
| 3. | Trials with two or more unclear site-EC matches | 75 (31.8%) |

## 42 trials in which the number of sites equaled the number of ECs, which was greater than one

There were 42 studies in which the number of sites was equal to the number of ECs, with each being greater than one (S9 File). In this set, there was often considerable difficulty in finding a site-EC match using a programme.

(i) A generic 'Institutional Ethics Committee' may have been repeated multiple times for a given trial. (ii) A site was repeated, and so too an EC. (iii) There were multiple instances where a site, with the same principal investigator (PI), was repeated. (iv) Alternatively, one site was listed twice, with different PIs. (v) There were several instances in which a given hospital chain listed several of its hospitals, in different cities, as sites. The ECs may or may not have listed the city, and therefore could not be matched to particular hospitals. (vi) In some studies, only a doctor's name was linked to the EC, and it was unclear which EC was linked to which site. (vii) A given trial had two sites that were represented by the same acronym. If the city was listed in one EC's field, then the other EC could also be assigned. (viii) There were variations in the name listed for the site and the EC. There were also spelling mistakes between the site and the EC. (ix) Abbreviations were used either in the site name or in the EC, making it more difficult to link a site and EC. (x) In one case a long and complicated acronym, listed as the 'site name', was not expanded anywhere in the trial record.

## One EC for multiple sites

On a separate note, we looked into the set of 53 trials with more sites than ECs and examined whether there were any in which a single EC was responsible for multiple sites. We identified only four studies that were registered on or after 25 March 2019. Of these, two cases (CTRI/2019/05/019077, CTRI/2020/01/022936) were registered retrospectively, with enrolment having started in 2012 and 2015, respectively. In the other two (CTRI/2019/03/018264, CTRI/2020/06/025800), the sites were in the same city as the EC.

In summary, we have described the problems with EC data and trying to determine the correct site-EC matches of 1359 studies, in various categories above. We illustrate these issues with specific examples in Table 5.

## Discussion

Public trial registries provide transparency around many aspects of trials, and this helps decision making in regard to healthcare. The mere listing of the name of an EC provides transparency around which entity has provided ethics clearance to a study. In fact, from the day it was established, CTRI has required trialists to provide details of the EC at the time of registering the study [22]. This was even before the World Health Organization (WHO) recommended that such information be available in public trial registries. Incomplete or incorrect data detracts from the tremendous potential benefit of such information in the database.

In this study, we wished to focus on Phase 2 and Phase 3 interventional trials. Although the specific EC approving a trial should have been settled by the time recruitment started, as a matter of abundant caution, we only included studies that were closed to recruitment or completed. It is possible that EC details were modified after the trial started, but the latest information would be reflected in the main record, and would be in our SQLite database after we downloaded the data.

Coming to the EC data itself, there are two angles from which we could, in principle, examine such information. First, as a field that provides details about the individual study. Second, as a field that we analyze across trials, to determine whether regulations have been followed, as

**Table 5. Examples of trials* that illustrate (i) various problems with the data in the EC fields of CTRI records, and (ii) challenges in identifying site-EC matches.**

| No. | Nature of problem or potential problem | CTRI ID and URL |
|---|---|---|
| | **Problem for analysis of the individual trial** | |
| 1 | A trial for which no EC was listed. | CTRI/2017/05/008477 |
| | | http://ctri.nic.in/Clinicaltrials/pmaindet2.php?trialid=13910 |
| 2 | A trial which listed what appeared to be a casual reference to an EC, rather than its proper name. | CTRI/2018/05/014249 |
| | | http://ctri.nic.in/Clinicaltrials/pmaindet2.php?trialid=22711 |
| 3 | A trial in which the EC fields only listed the names of the sites. | CTRI/2020/05/025254 |
| | | http://ctri.nic.in/Clinicaltrials/pmaindet2.php?trialid=43630 |
| 4 | A trial with an EC acronym not spelled out anywhere in the record. | CTRI/2019/10/021480 |
| | | http://ctri.nic.in/Clinicaltrials/pmaindet2.php?trialid=37023 |
| 5 | A trial with one generically named 'Institutional Ethics Committee' for multiple sites. (in the example provided, there were sites in four parts of the country, but only one Institutional Ethics Committee was listed.) | CTRI/2019/05/019077 |
| | | http://ctri.nic.in/Clinicaltrials/pmaindet2.php?trialid=22408 |
| 6 | A trial in which a generically named 'Institutional Ethics Committee' was listed multiple times. (in the example provided, there were sites in four parts of the country, but the term Institutional Ethics Committee was listed three times.) | CTRI/2016/06/007016 |
| | | http://ctri.nic.in/Clinicaltrials/pmaindet2.php?trialid=12531 |
| 7 | A trial with one site, but ECs from three organizations. It was unclear which of the ECs was, or were, linked to the site. | CTRI/2017/08/009296 |
| | | http://ctri.nic.in/Clinicaltrials/pmaindet2.php?trialid=17839 |
| 8 | A trial with many ECs. In almost all the cases, there was a doctor's name in the EC that could be linked to the PI's name linked to the site. However in one case, a different doctor's name was listed in the site and in the EC. | CTRI/2018/02/012059 |
| | | http://ctri.nic.in/Clinicaltrials/pmaindet2.php?trialid=20815 |
| 9 | One hospital chain listed 2 of its hospitals, in different cities, as sites. The ECs did not list the cities, and therefore could not be matched to particular hospitals. | CTRI/2019/12/022455 |
| | | http://ctri.nic.in/Clinicaltrials/pmaindet2.php?trialid=38234 |
| 10 | Five trials that listed the same EC file number. | CTRI/2019/01/017109 |
| | | http://ctri.nic.in/Clinicaltrials/pmaindet2.php?trialid=30219 |
| | | CTRI/2019/01/017259 |
| | | http://ctri.nic.in/Clinicaltrials/pmaindet2.php?trialid=30484 |
| | | CTRI/2019/01/017261 |
| | | http://ctri.nic.in/Clinicaltrials/pmaindet2.php?trialid=30703 |
| | | CTRI/2019/01/017327 |
| | | http://ctri.nic.in/Clinicaltrials/pmaindet2.php?trialid=30381 |
| | | CTRI/2019/01/017338 |
| | | http://ctri.nic.in/Clinicaltrials/pmaindet2.php?trialid=30700 |
| | **Problem for data analytics** | |
| 1 | A trial with a single site, that listed the EC thrice. | CTRI/2020/07/026337 |

*(Continued)*

**Table 5.** (Continued)

| No. | Nature of problem or potential problem | CTRI ID and URL |
|---|---|---|
| | | http://ctri.nic.in/Clinicaltrials/pmaindet2.php?trialid=44863 |
| 2 | A trial with multiple sites, in which 3 Institutional ECs approved one site. | CTRI/2019/06/019509 |
| | | http://ctri.nic.in/Clinicaltrials/pmaindet2.php?trialid=29960 |
| 3 | A trial with one site, but ECs from three organizations. One of the ECs was clearly linked to the site. | CTRI/2020/05/025209 |
| | | http://ctri.nic.in/Clinicaltrials/pmaindet2.php?trialid=43251 |
| 4 | A trial in which a site was duplicated in order to list two ECs. The ECs belong to (a) the institution running the study and (b) the Council to which the institution belonged. | CTRI/2018/05/013636 |
| | | http://ctri.nic.in/Clinicaltrials/pmaindet2.php?trialid=24764 |
| 5 | A trial with 39 sites, but a single EC (which was implied, because the organization name was listed in the EC field.) | CTRI/2018/03/012804 |
| | | http://ctri.nic.in/Clinicaltrials/pmaindet2.php?trialid=19928 |
| 6 | A trial in which there was a spelling mistake in the acronym used for the EC, creating a difference between the site name and the acronym. | CTRI/2016/11/007473 |
| | | http://ctri.nic.in/Clinicaltrials/pmaindet2.php?trialid=15321 |
| 7 | An acronym used in the EC, but not in the site name. | CTRI/2017/05/008638 |
| | | http://ctri.nic.in/Clinicaltrials/pmaindet2.php?trialid=18581 |
| 8 | A trial in which, although it appeared that two ECs were listed, one of them was just a message. | CTRI/2019/08/020756 |
| | | http://ctri.nic.in/Clinicaltrials/pmaindet2.php?trialid=20800 |
| 9 | A trial with two sites having the same acronym in the EC. The correct site-EC match could be established only because one of the ECs also listed the city. | CTRI/2016/06/007062 |
| | | http://ctri.nic.in/Clinicaltrials/pmaindet2.php?trialid=14511 |
| 10 | Three trials, between them, illustrated the use of the same acronym (KIMS) for 4 organizations: Kerala Institute of Medical Sciences and Karnataka Institute of Medical Sciences Kempegowda Institute Of Medical Sciences Hospital Kalinga Institute of Medical Sciences | CTRI/2016/06/007062 |
| | | http://ctri.nic.in/Clinicaltrials/pmaindet2.php?trialid=14511 |
| | | CTRI/2019/12/022312 |
| | | http://ctri.nic.in/Clinicaltrials/pmaindet2.php?trialid=36915 |
| | | CTRI/2019/04/018384 |
| | | http://ctri.nic.in/Clinicaltrials/pmaindet2.php?trialid=31244 |
| 11 | A trial for which a long and complicated acronym represented the site name. The acronym was not explained anywhere in the trial record. | CTRI/2020/08/027163 |
| | | http://ctri.nic.in/Clinicaltrials/pmaindet2.php?trialid=46406 |
| 12 | A trial in which the site was listed twice, with the same PI. The correct EC was listed once. | CTRI/2018/05/014015 |

(*Continued*)

**Table 5.** (Continued)

| No. | Nature of problem or potential problem | CTRI ID and URL |
|---|---|---|
| | | http://ctri.nic.in/Clinicaltrials/pmaindet2.php?trialid=24470 |
| 13 | A trial in which the site was listed twice, with different PIs. The correct EC was listed once. | CTRI/2019/03/018196 |
| | | http://ctri.nic.in/Clinicaltrials/pmaindet2.php?trialid=31843 |
| 14 | A trial for which the name of the site was not provided anywhere in the Sites of Study field, although present elsewhere in the document. | CTRI/2018/05/014268 |
| | | http://ctri.nic.in/Clinicaltrials/pmaindet2.php?trialid=23824 |
| 15 | A trial in which there were differences in the spelling of the name of the site and the reference to the site in the EC field. | CTRI/2020/04/024749 |
| | | http://ctri.nic.in/Clinicaltrials/pmaindet2.php?trialid=42972 |
| 16 | A trial in which, for many sites, the relevant EC had a doctor's name linked to it. The EC could be linked to the site only by referencing the PI field. | CTRI/2018/02/012059 |
| | | http://ctri.nic.in/Clinicaltrials/pmaindet2.php?trialid=20815 |
| 17 | A trial that had multiple sites with the generic name 'Government Medical College and Hospitals'. The EC could be linked to the site only by referencing the site address field. | CTRI/2017/06/008843 |
| | | http://ctri.nic.in/Clinicaltrials/pmaindet2.php?trialid=15620 |
| 18 | A trial for which an independent EC was not identified as such. | CTRI/2018/08/015547 |
| | | http://ctri.nic.in/Clinicaltrials/pmaindet2.php?trialid=27873 |

*To be noted, 'trial' means 'trial record'.

inputs to policy making, and so on. For the latter use, a data-analytics approach is required, for which data needs to be in a suitable format. The CTRI template captures data in a systematic way. Nevertheless, incomplete or ambiguous data is likely to complicate–or prevent–certain analyses. In this study, we found that many EC names were indeed listed in a sufficiently detailed and unambiguous manner. For example, a trial with three Government Medical Colleges, listed the city in both the site name and in the EC name, and therefore could be readily matched. However, the details in other cases were problematic. Here we discuss the problems identified above, and then comment more generally.

It is surprising that three interventional trials, with Phase 2/Phase 3; Phase 3; or Phase 3/Phase 4, respectively, do not list an EC. It is possible that the Phase listed was incorrect, since one case was also listed as a Behavioral study. Or, an EC may have cleared the trial, but the EC information was not entered in the registry. In earlier work that was primarily concerned with other parts of the study record, we urged the managers of CTRI to implement logic rules to prevent internal inconsistencies in a record [9]. To prevent the error under discussion, a logic rule could be implemented that would prevent the record of an interventional trial being submitted if the EC field was left blank. We see this as an important step in helping strengthen the data in this field.

There also needs to be a more intensive quality check of a submission prior to it being allowed to be published. Although automated checks would be the most efficient, it is essential that a CTRI staffer review the record in a detailed manner, to minimize errors and ensure that

the data are meaningful. Although this is happening already [23], it clearly needs to be strengthened.

We now come to some of the problems with the details of the EC.

a. When the name of the EC is not listed, and only the institution is named, that is equivalent to leaving the EC field blank. In the case of either a generic name of an EC, or what appears to be a casual reference to an EC, it is unclear whether this is the formal name of the EC. An earlier study found that PIs from a given institution in India may have registered a particular EC with 10 different 'names' [24]. If the correct EC name is not listed, then the field is again essentially unfilled. However, the presence of an incorrect name will confound attempts to analyze the records of large numbers of trials.

b. It is possible that the data where a single EC was listed for 'n' sites were correct, but it was impossible to determine this. Ideally, the EC should have been repeated 'n' times, with further details to identify it unambiguously. Or, there should have been some other way to indicate that this was the only committee for all the sites. This study has found that many trials had errors or ambiguities in their listed ECs. As such, merely listing one EC for multiple sites does not settle the matter of how many ECs cleared the study. Separately, in trials with multiple sites and ECs, some sites did not appear to have EC partners, and it is conceivable that several sites were linked to a few ECs. However, it was impossible to determine whether this was so, and if so, to identify which EC was linked to which site. The same situation holds where a generically named Institutional Ethics Committee is listed for multiple sites–it was impossible to determine which EC it was, and which site(s) it was linked to. Going forward, if an EC is repeated, a logic rule should flag such a record. If the trialist confirms that the repetition of the EC is correct, then the record could be accepted.

In the 53 trials that had more sites than ECs, we found that the two studies that were registered prospectively on or after 25 March 2019 were compliant with the rule that required the EC to be located within the same city as, or within 50 km of, the trial site. If the EC is able to discharge its duties better with such restrictions on its location, then the value of this aspect of the New Drugs and Clinical Trials Rules, should have impact in trials registered after the rule took effect.

c. We identified 13 trials that listed what appeared to be the EC file numbers. The case of five trials having the same file number was inexplicable. This is irrelevant information, that should not be submitted in the EC field.

d. When a given institution's EC was represented by different names, it was unclear whether this was a reference to the same EC or not. Since we encountered a study where a given site was approved by three generically named ECs, IEC I, II and III, in principle other IECs may not have been unique for a given institution. There is also an earlier report of hospitals with multiple ECs [19]. Accordingly, it is important that the ECs be listed in an unambiguous manner. To be noted, as far as the trial record goes, in case only one EC needs to have cleared a study, two need not be listed. Potentially, this could lead to confusion, especially in any automated analysis.

e. We now come to the issue of independent ECs. Some independent ECs were identified as such, but all of them were not. In North America, such ECs have been criticized because of the perception that they treat trial sponsors as clients, and a source of revenue [25]. Some independent ECs in India have been opaque about details of the committee, and have also charged steep fees to clear studies. This has given rise to the charge that such ECs seemed to be more focused on generating 'ethic review business' than on protecting the rights of trial

participants [26]. Some of these independent ECs have been involved with trials that became controversial due to multiple ethics violations [27]. In view of some of these issues, as far as regulatory trials go, in 2013 the Government of India restricted independent ECs' review to Bioavailability and Bioequivalence studies [28], although this was reversed in 2019 [21]. Given the criticism of independent ECs, it would be helpful to identify such ECs, to enable separate analysis. Possibly, in the registration form, the trialist should be required to classify each committee as an independent or institutional EC. To be noted, although the non-identification of independent committees has been listed in Table 5 along with problematic cases, this is more of a sub-optimal than a problematic situation.

After discussing some of the problems with the EC data, we now come to some of the problems with the details of the trial sites, and to some of the other challenges in establishing site-EC linkages.

i. We first consider the listing of more ECs than sites. The CTRI staff have explained one possible cause of this [29], as follows. If new sites are to be added to the CTRI record, the trialist first uploads the ethics approval documents. Then the site field is unlocked and the new site information is added. There may be a lag between the two steps, leading to an excess of ECs over sites for the intervening period. We suggest that (a) the public should not see each step of the process, and only get to see the final changes, or (b) there should be some indication that updating the record with new EC and site information is ongoing. Another possible reason for more ECs is that sites may have been dropped, without the corresponding changes in the list of ECs. Other situations may be as follows. For particular trials, one EC may seek the opinion of another EC before approving a trial. Conceivably, this may lead to more than one EC per site being listed for that trial. Also, in 'multicentre review' the EC of one institution may approve and oversee the trial at all the sites, with the local EC also involved in overseeing the trial. However this would happen in low risk non-regulatory trials, and would not be possible in the Phase 2 and Phase 3 regulatory trials of this study. If the number of sites and number of ECs is different, there ought to be a logic rule that flags the record. It should not be possible to submit a record, or an amendment to the record, unless this number is the same, or the difference is explained.

ii. Differences in spelling between site and EC, incorrect spellings, and different representations of a given institution's name make it hard to match site and EC, especially if one wishes to automate the process. Likewise, if the *Name of Site* does not contain the correct name of the site, and only the name of the department, for instance, then any attempt to link the EC to a site becomes complicated. As suggested in our earlier work, if organizations' names were pre-registered, and then selected from a drop-down menu while registering a trial, this would prevent this category of problems [9].

iii. It is an outright error if a site is repeated, with the same or different PIs. Conceivably, a logic rule could check whether a site is repeated, and not permit the submission of such a record.

iv. Some of the challenges, such as (a) multiple sites of a hospital chain, and (b) multiple instances of a generic site name such as Government Medical College, might be resolved by referring to the *Site Address*. Likewise, in cases where the EC has a generic name such as IEC, but also lists a doctor's name in the EC field, this name may be available in the list of PIs, and therefore provide a clear link to a site. However, for the purpose of data analytics, the use of another field, with entries that may also be non-standard or idiosyncratic, will be non-trivial. Ideally, one should not have to refer to other fields to establish the site-EC linkage.

v. We now come to the use of acronyms. We encountered several instances of two or more organizations sharing the same acronym. Also, acronyms were sometimes created in an idiosyncratic manner. Acronyms that are not explained elsewhere in the study record are another challenge. Further, the incorrect spelling of an acronym could lead to confusion, if the mis-spelled acronym was the acronym for another organization. In general, the use of acronyms is likely to create ambiguity, for a data analytics approach in particular.

Having discussed some of the problems identified with the EC or site data, we now make some more general points.

Although we did not attempt to quantify the challenge of linking sites to ECs for all the trials of this study, we did this for two subsets, comprising 16 and 236 trials, which together constituted 18.5% of the 1359 trials of this study. In these two sets, 50% and 60% of the trials, respectively, had sites without clear links to an EC. As such, a very significant fraction of trials had one or more sites for which the EC could not be readily assigned. To be noted, this 'matching' of EC with site does not identify the EC itself unambiguously. It merely indicates that an EC from the relevant institution is overseeing the trial at that institution. A given institution is permitted to have multiple ECs, and bigger hospitals tend to do so. For instance, in this study, we encountered (a) Institutional Ethics Committee (IEC-I) and (IEC-II) were involved in one of the sites of the trial CTRI/2017/08/009485 and (b) IEC-I,II,III- TATA Memorial Hospital were involved in one of the sites of the trial CTRI/2019/06/019509. We note that in each case, this listing of multiple ECs was in a single cell, so it did not complicate the issue of how many ECs approve a given site when one tries to match the 'EC cell' with the 'site cell'. Therefore, the mere listing of 'EC' or 'IEC', in full form or as an acronym, does not clarify whether it is a uniquely identifiable committee of the institution.

For several of the issues mentioned above, such as the listing of an 'EC' which was just a message, or the duplication of a site or EC, the sponsor or trialist was at fault for entering incorrect information. However, the CTRI managers do inspect each record at the time of registration, and return it to the sponsor for correction if needed. This provides an opportunity to catch mistakes, especially if these mistakes occurred during the initial filing of the study record. Although this is already happening, clearly the process needs to be strengthened.

From the point of view of our analysis, the best trials were the four that listed the EC registration number. Since these were single site and single EC studies, and no change in the EC information had been recorded over time for any of them, the EC was almost certainly linked to that site. However, in principle, a trial that initially had multiple sites and multiple ECs may have subsequently been reduced to a single site and a single EC. In such a case, a mismatch error between the site and the EC could have arisen, and even listing the EC registration number would not confirm which EC was linked to the site. Nevertheless, over time, there may be further improvements in the data held in the CTRI records, and this will generate greater confidence that the EC registration number linked to a site is the correct EC. As such, the listing of the registration number should be made mandatory for each EC linked to a trial.

An earlier study found that many ECs were not registered with CDSCO as they were supposed to be [24]. If the listing of the EC registration number is made mandatory, then it will also tackle the issue of trials being approved by ECs that are not registered with CDSCO for regulatory trials, or the Department of Health Research, Ministry of Health and Family Welfare, Government of India for other clinical research. A logic rule should prevent a trial from being accepted if valid EC registration numbers are not provided for all sites. It is possible that after the trial begins, the EC renews its registration and receives a new number. This could lead to confusion, and therefore we believe that the EC registration number should remain the

same but the year of registration should change. Further, there should be a record of all relevant registration numbers of the EC in the EC field.

CTRI is a unique and invaluable resource, of relevance to every clinical trial ever run in the country since the establishment of the registry. Regulatory decisions will be made based on data in this registry. Medical professionals, the public, philanthropic organizations and others will also make decisions based on this data. It is of utmost importance that the registry has the best possible records. Based on a range of parameters, a recent study ranked CTRI eleventh out of the 18 major public trial registries that included ClinicalTrials.gov of the United States and the 17 registries that WHO considers Primary Registries [1]. The EC field was only a minor part of the assessment, since registries were merely scored for the presence or absence of the EC field. Therefore, CTRI scored fine for this field. However, this study has identified several lacunae in EC or site-EC linkage data.

Over the years, researchers have pointed out the many steps that need to be taken to improve the ethics of clinical trials in India, including more training of EC members [27], increased active monitoring of ongoing trials [24], and the need for more ECs to be accredited [27], including by international organizations [24]. Our focus has been the quality of CTRI records. In earlier work [9], we identified problems that affect many fields in a CTRI record. Of the many deficiencies documented, a few concerned the EC. Here, we have extended that work in various ways, and have identified many other types of errors or ambiguities (Table 5).

We end by making a few recommendations to improve the EC-related data in CTRI. As suggested in our earlier work (a) the table listing the number of sites and the table listing the ethics committees could be merged such that the EC linked to a given site is clear, and (b) if the site name were pre-registered, and chosen from a drop-down menu while registering the trial, this would prevent inconsistencies and ambiguity in the identity of the site. Additionally, and most importantly, the EC registration number must be provided for each EC, and a trial should not be registered without it. We also believe that the implementation of several logic rules will prevent some of the errors or ambiguities identified in this study. Finally, there also needs to be a more intensive quality check of a submission prior to it being allowed to be published. These steps will raise user confidence in CTRI EC data, and enable data-based public policy and inferences. This will also contribute to increased transparency, and trust, in clinical trials, and their oversight, in India.

## Conclusions

In this study, we have examined the site and EC data for 1359 Phase 2 or Phase 3 interventional trials registered with CTRI after 1 June 2016. We have identified a large number of problems with the data in the EC or site field, that create a challenge to link particular sites with particular ECs, especially if a programme is used to find the matches. We make a few suggestions on how the situation could be improved. The four most important suggestions are to list the EC registration number for each EC, to merge the site and EC tables so that it is clear which EC is linked to which site, to implement various logic rules that would prevent a trial from being registered unless certain conditions were met, and to strengthen the inspection of each record before it is published. Users would have even greater confidence in CTRI data if such steps were taken.

## Supporting information

**S1 File. A sample CTRI record.**
(PDF)

**S2 File. The R script used to download data from CTRI, process them and store them in an SQLite database.**
(DOCX)

**S3 File. Details of 31,423 records from CTRI, stored in an SQLite database.**
(DOCX)

**S4 File. The schema of the SQLite database.**
(XLS)

**S5 File. Expanded methods.**
(DOC)

**S6 File. The steps taken to identify all trials, registered with CTRI between 1 June 2016 and 19 February 2021, that were interventional, Phase 2 or Phase 3, with completed recruitment or closed to recruitment.**
(XLSX)

**S7 File. The steps taken to identify the set of 1359 unique Phase 2 and Phase 3 trials with either single or multiple sites each.**
(XLSX)

**S8 File.** **A.** The breakup of the set of 1359 trials into those with (i) no EC, (ii) more sites than ECs, (iii) more ECs than sites, (iv) an equal number of sites and ECs. **B.** For two of these subsets, a listing of the details of the site name and ECs, and for one subset, statistics of sites without an EC partner.
(XLSX)

**S9 File.** **A.** The breakup of the set of 1265 trials for which the number of sites equaled the number of ECs into those with (i) single sites, (ii) more than one site (simple cases) and (iii) more than one site (complicated cases). **B.** For each subset, a listing of the details of the site name and ECs, and for one subset, statistics of sites without an EC partner.
(XLSX)

## Acknowledgments

We thank Dr. Sangeeta Kumari for discussions.

## Author Contributions

**Conceptualization:** Gayatri Saberwal.

**Data curation:** Indraneel Chakraborty.

**Formal analysis:** Indraneel Chakraborty, Adya Shreya, Anant Bhan, Gayatri Saberwal.

**Funding acquisition:** Gayatri Saberwal.

**Investigation:** Indraneel Chakraborty, Adya Shreya, Gayatri Saberwal.

**Methodology:** Indraneel Chakraborty, Gayatri Saberwal.

**Project administration:** Gayatri Saberwal.

**Resources:** Gayatri Saberwal.

**Software:** Indraneel Chakraborty.

**Supervision:** Gayatri Saberwal.

**Validation:** Jaishree Mendiratta, Gayatri Saberwal.

**Visualization:** Indraneel Chakraborty, Adya Shreya, Gayatri Saberwal.

**Writing – original draft:** Gayatri Saberwal.

**Writing – review & editing:** Indraneel Chakraborty, Adya Shreya, Jaishree Mendiratta, Anant Bhan, Gayatri Saberwal.

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
