## [Decision Letter · Decision Letter 0]

13 Apr 2022

PGPH-D-21-00390

An analysis of deficiencies in the Ethics Committee data of interventional trials registered with the Clinical Trials Registry – India

Dear Dr. Saberwal,

Thank you for submitting your manuscript to PLOS Global Public Health. After careful consideration, we feel that it has merit but does not fully meet PLOS Global Public Health’s publication criteria as it currently stands. Therefore, we invite you to submit a revised version of the manuscript that addresses the points raised during the review process.

We look forward to receiving your revised manuscript.

Kind regards,

Julia Robinson

Executive Editor

Journal Requirements:

1. Please update the completed 'Competing Interests' statement. Please declare all competing interests beginning with the statement “I have read the journal's policy and the authors of this manuscript have the following competing interests:”.

Additional Editor Comments (if provided):

Reviewers' comments:

Reviewer's Responses to Questions

**Comments to the Author**

1. Does this manuscript meet PLOS Global Public Health’s publication criteria? Is the manuscript technically sound, and do the data support the conclusions? The manuscript must describe methodologically and ethically rigorous research with conclusions that are appropriately drawn based on the data presented.

Reviewer #1: Yes

Reviewer #2: No

2. Has the statistical analysis been performed appropriately and rigorously?

Reviewer #1: Yes

Reviewer #2: No

3. Have the authors made all data underlying the findings in their manuscript fully available (please refer to the Data Availability Statement at the start of the manuscript PDF file)?

Reviewer #1: Yes

Reviewer #2: Yes

4. Is the manuscript presented in an intelligible fashion and written in standard English?

Reviewer #1: Yes

Reviewer #2: No

5. Review Comments to the Author

Reviewer #1: This is a well-written article that describes specific categories of problems that prevent the clear identification of ethics committees that have approved a clinical trial at a given site. The authors argue that this information is important to ensure transparency of trial registries and that this is ultimately important to build public trust. The main arguments and findings are clear, but I have a few questions and clarifications I would like the authors to consider.

“Ethics Committee (EC)” – it might be better to change EC to REC to distinguish research ethics committees from other kinds of ethics committees, such as clinical ethics committees. I recommend that the same change is made in the key words in order to improve the accuracy of key word searches. I do, however, also take note of the following sentence in the discussion: “Even leaving aside a high clinical burden, it is known that ECs may be overburdened with trial-related work”. Does this mean that ECs act as both research ethics committees and clinical committees in India? If so, could this please be described in the Introduction section in order to provide readers with more context of the ethics review system in India?

In addition, could the authors provide more information about the range of trials captured on the Clinical Trials Registry–India.

Registries certainly have many uses and the authors identify a number: identifying gaps and errors in registries or between registries; identifying discrepancies between the data in registries and in publications; investigating whether trial results were reported on time”. From this description, it is however not clear what value it has “for the public to determine which committee gave approval to a given study at a given site”. While more justification is provided in the discussion, it would be more informative if a clear rationale could be provided upfront since this is the main focus of the article.

The title refers to “interventional trials”, but the introduction refers to “clinical trials”. The authors state: “we went on to identify the trials of interest in this set”. Could they please explain why they were only interested in interventional trials and not all clinical trials?

There were 24,445 trials registered between 1 June 2016 and 19 February 2021, an almost five-year period”. Why was this specific time period selected? Why not rather exactly 5 years?

“17,342 were interventional trials, as identified by the field Type of trial. Of these, 3160 were Phase 2, and 3200 were Phase 3, as identified by the field Phase”. This leaves a large number of trials (a total of 10,982) that were not Phase 2 and/ or 3. Please explain what these trials were.

Why were Phase 1 trials excluded?

A total of 1576 trials were identified but only 1359 were used. Were all the others duplicates? If so, why would there be such a large number of duplicates in the registry?

“For the unique set of 1359 trials, we went on to identify all those for which (i) no EC was listed; (ii) the number of sites exceeded the number of ECs; (iii) the number of ECs exceeded the number of sites; and (iv) the number of sites was the same as the number of ECs.” For readers to make sense of this analysis, can the authors please briefly explain the ethics review system in India. For instance, does each research site have its own research ethics committee? Are some research ethics committees able to approve more than one research site? Are academic researchers expected to obtain additional ethics approval from the research ethics committee of their own institution if the research site is not affiliated to their institution? While some such relevant information is provided in line 320 onwards, it would be helpful for readers to have more context upfront.

“For many of the 1359 trials, EC names were listed in a sufficiently detailed manner.” Please quantify “many”.

“(ii) Generic names of the EC were used quite often. The name of the institution may, or may not, have been listed in this field as well. (iii) Some entries appeared to be casual references to particular ECs, rather than their proper names.” What is the difference between “generic names” and “casual references”?

Table 5: “To be noted, ‘trial’ means ‘trial record’” should not be part of the table title but should rather be added as a footnote.

It is unclear how 5 and 6 are different:

5. “A trial with one generically named ‘Institutional Ethics Committee’ for multiple sites.

6 A trial in which a generically named ‘Institutional Ethics Committee’ was listed multiple times.

Can the authors please explain what they mean by the following: “Although clarification should not be necessary, this study has found that many trials had errors or ambiguities in their listed ECs.” Why would clarification not be necessary?

“Some of those that were registered before this date were not, which highlights the importance of such a rule.” It is not clear to me why trials registered before the rule came into effect should have been compliant with the rule.

“It is possible that after the trial begins, the EC renews its registration and receives a new number.” Would it not be sensible to recommend that research ethics committee registration numbers remain the same but that the year of registration is indicated?

“Even leaving aside a high clinical burden, it is known that ECs may be overburdened with trial-related work”. Does this mean that ECs act as both research ethics committees and clinical committees?

“Based on a range of parameters, a recent study ranked CTRI eleventh out of the 18 major public trial registries”. Please clarify where these registries are located.

Small issues:

“registry data has” - the word data is plural.

“the trials enterprise” – should this be the clinical trials enterprise?

“Repeatedly, academics and others, including the Science and Technology Committee of the UK House of Commons [3–5]…”. Since only reference 5 refers to the Science and Technology Committee of the UK House of Commons, it would be better to place the references as follows: Repeatedly, academics and others [3,4], including the Science and Technology Committee of the UK House of Commons [5]...

Reviewer #2: Detailed Comments attached. This research has been conducted without any ethics committee approval or without seeking an exemption from ethics committee before undertaking this analysis or without registering the study on CTRI platform before its conduct. Further this paper is based on an interpretation of findings and not on verified facts/ discussion regarding data in consultation with data managers. This is not a true presentation of facts and not recommended for publication in this esteemed journal.

6. PLOS authors have the option to publish the peer review history of their article (what does this mean?). If published, this will include your full peer review and any attached files.

**Do you want your identity to be public for this peer review?** For information about this choice, including consent withdrawal, please see our Privacy Policy.

Reviewer #1: No

Reviewer #2: No

---

## [Decision Letter · Decision Letter 1]

15 Aug 2022

PGPH-D-21-00390R1

An analysis of deficiencies in the Ethics Committee data of certain interventional trials registered with the Clinical Trials Registry – India

Dear Dr. Saberwal,

Thank you for submitting your manuscript to PLOS Global Public Health. After careful consideration, we feel that it has merit but does not fully meet PLOS Global Public Health’s publication criteria as it currently stands. Therefore, we invite you to submit a revised version of the manuscript that addresses the points raised during the review process.

In this new round of examination of the manuscript we found two conflicting opinions. After carefully analyzing the arguments of the two reviewers, I disagree with the reasons given for the rejection of the article and I agree with the suggestions that were made so that the article can be accepted with minor revisions: 1) a brief description of the steps required by local regulations for the registration and ethical evaluation of projects until their approval for initiation are described; and 2) the inclusion of the meaning of the acronym "BA/BE studies".

We look forward to receiving your revised manuscript.

Kind regards,

Deisy de Freitas Lima Ventura, Ph.D

Section Editor

Journal Requirements:

Reviewers' comments:

Reviewer's Responses to Questions

**Comments to the Author**

1. If the authors have adequately addressed your comments raised in a previous round of review and you feel that this manuscript is now acceptable for publication, you may indicate that here to bypass the “Comments to the Author” section, enter your conflict of interest statement in the “Confidential to Editor” section, and submit your "Accept" recommendation.

Reviewer #2: (No Response)

Reviewer #3: All comments have been addressed

2. Does this manuscript meet PLOS Global Public Health’s publication criteria? Is the manuscript technically sound, and do the data support the conclusions? The manuscript must describe methodologically and ethically rigorous research with conclusions that are appropriately drawn based on the data presented.

Reviewer #2: No

Reviewer #3: Yes

3. Has the statistical analysis been performed appropriately and rigorously?

Reviewer #2: No

Reviewer #3: N/A

4. Have the authors made all data underlying the findings in their manuscript fully available (please refer to the Data Availability Statement at the start of the manuscript PDF file)?

Reviewer #2: Yes

Reviewer #3: Yes

5. Is the manuscript presented in an intelligible fashion and written in standard English?

Reviewer #2: No

Reviewer #3: Yes

6. Review Comments to the Author

Reviewer #2: (No Response)

Reviewer #3: The article is interesting and addresses an issue that is very important in the entire process of ethical clearance of health research projects: transparency and accountability.

In order to make the Indian context better understood for the reader of the article who is not familiar with it, I suggest that, very briefly, the steps required by local regulations for the registration and ethical assessment of projects until their approval for the start are described.

Although the authors were careful to explain the meaning of each acronym used in the text, the same was not done for “BA/BE studies”.

7. PLOS authors have the option to publish the peer review history of their article (what does this mean?). If published, this will include your full peer review and any attached files.

**Do you want your identity to be public for this peer review?** For information about this choice, including consent withdrawal, please see our Privacy Policy.

Reviewer #2: No

Reviewer #3: **Yes: **Sergio Rego

---

## [Editor Report · Decision Letter 2]

28 Sep 2022

An analysis of deficiencies in the Ethics Committee data of certain interventional trials registered with the Clinical Trials Registry – India

PGPH-D-21-00390R2

Dear Prof. Saberwal,

We are pleased to inform you that your manuscript 'An analysis of deficiencies in the Ethics Committee data of certain interventional trials registered with the Clinical Trials Registry – India' has been provisionally accepted for publication in PLOS Global Public Health.

Best regards,

Julia Robinson, on behalf of Deisy de Freitas Lima Ventura (Section Editor)